# Gemcitabine–Paclitaxel Chemotherapy for Patients with Advanced Urothelial Cancer Refractory to Cisplatin-Based Chemotherapy: Predictive Role of PGK1 for Treatment Response to Cytotoxic Chemotherapy

**DOI:** 10.3390/ijms232012119

**Published:** 2022-10-11

**Authors:** Dai Koguchi, Kazumasa Matsumoto, Masaomi Ikeda, Yuriko Shimizu, Marie Nakamura, Yutaka Shiono, Hiroki Katsumata, Yuichi Sato, Masatsugu Iwamura

**Affiliations:** Department of Urology, Kitasato University School of Medicine, 1-15-1 Kitasato Minami-ku, Sagamihara 252-0374, Japan

**Keywords:** cancer of the urinary tract, gemcitabine, paclitaxel, survival, biomarkers

## Abstract

An investigation of alternatives to immune checkpoint inhibitors for advanced urothelial cancer (aUC), with biologic information, is urgently needed. Clinical data for 53 patients who received gemcitabine–paclitaxel therapy (GP) as 2nd-line chemotherapy for aUC refractory to platinum-based chemotherapy were retrospectively reviewed. The efficacy and tolerability of GP were evaluated, and the predictive value of phosphoglycerate kinase 1 (PGK1) immunostained in surgical specimens was investigated for treatment outcomes in 1st- and 2nd-line chemotherapy. GP was associated with an objective response rate of 35.8% and a median overall survival duration of 12.3 months. Multivariate analysis showed that PS2 and 1st- and 2nd-line non-response are independent predictors of worse progression-free survival and that PS2 and 1st-line non-response are independent predictors of worse overall survival. Adverse events were manageable, and no therapy-related deaths occurred. Non-response rates to 1st-line chemotherapy were significantly higher in patients with a high expression of PGK1 in the nucleus than in those with low expression (*p* = 0.006). Our study demonstrates the efficacy and tolerability of 2nd-line GP for patients with aUC who are refractory to platinum-based chemotherapy. Moreover, PGK1 in the nucleus was predictive values for resistance to platinum-based chemotherapy in aUC.

## 1. Introduction

Urothelial cancer (UC) can be located in either or both of the bladder and upper urinary tract. Muscle-invasive disease is found at initial diagnosis in about 30% of patients with bladder cancer (BCa) and in about 60% of those with upper tract UC (UTUC), even though UTUC rarely accounts for more than 10% of all UCs [1,2]. Radical surgery such as cystectomy and nephroureterectomy has been the standard of care for muscle-invasive UC, and yet about half the patients undergoing radical surgery have lethal metastatic disease [3]. Moreover, 20% of muscle-invasive UC is initially diagnosed as metastatic or unresectable and is associated with a poor 5-year survival rate of <5% [3]. Given the aggressive nature of advanced UC (aUC), management of this disease has thus been a clinical challenge for decades.

Historically, UC has long been recognized as a chemosensitive disease, and platinum-based chemotherapy has been the most common 1st-line therapy for patients with aUC. Still, overall survival (OS) in patients with aUC treated with platinum-based chemotherapy is reported to be only 14 months [4]. Various 2nd-line chemotherapy agents have therefore been examined, and pembrolizumab, an immune checkpoint inhibitor (ICI) targeting PD-1, has been widely accepted as the standard 2nd-line treatment since the KEYNOTE-045 trial [5]. However, the overall response rate (ORR) to this ICI remains low (about 30% at maximum), and severe immune-related adverse events reportedly occur in about 15% of patients receiving it [5,6]. These therapeutic concerns highlight the need to continue exploring the efficacy of 2nd-line cytotoxic chemotherapies (CCs) for patients with platinum-refractory UC. In 2001, Meluch et al. were the first to report the efficacy of gemcitabine–paclitaxel (GP) as 1st-line therapy in patients with aUC [7]. Since 2005, GP has been the standard 2nd-line treatment in patients with aUC at Kitasato University Hospital who had previously been treated with platinum-based chemotherapy, being associated with an ORR of 42% and a median OS of 12.4 months [8,9]. Others also demonstrated acceptable outcomes with GP in platinum-refractory aUC [10,11,12,13]. However, data about the 2nd-line use of GP in aUC are scarce, especially since the advent of ICIs. Hence, using updated data, we aimed to retrospectively evaluate the efficacy and tolerability of GP as a subsequent treatment in patients who had previously received platinum-based chemotherapy for aUC.

Additionally, the most effective use of CC in aUC, an investigation of reliable biomarkers that will predict the response to CC is urgently needed. We therefore focused on phosphoglycerate kinase 1 (PGK1), a family of glycolytic enzymes that catalyze the reversible conversion of 1,2-bisphosphoglycerate to 3-phosphoglycerate [14]. We previously found high levels of PGK1 expression in the cisplatin-resistant T24 cell line, but other data about PGK1 in UC are limited to findings from a few gene enrichment analyses that simply show poor OS in patients with PGK1 overexpression [15,16,17]. In the present study, we therefore also investigated the association between PGK1 expression in UC specimens and the outcomes of 1st- and 2nd-line chemotherapy for aUC.

## 2. Results

### 2.1. Patient Characteristics

Table 1 shows the characteristics of the patients who received GP. The study cohort consisted of 42 men (79.2%) and 11 women (20.8%) with a median age of 68 years when they received GP. Of those 53 patients, 32 (60.4%) had BCa and 21 (39.6%) had UTUC. In terms of their prior chemotherapy regimen, 45 had received MVAC (84.9%) and 8 had received GC (15.1%) for a median of 4 cycles during both regimens (range: 2–8).

Patients in this study cohort received at least two cycles of GP, and the median number of cycles overall was four (range: 2–12). The median weekly dose intensity was 750 mg/m^2^ for gemcitabine (range: 500–1000 mg/m^2^) and 150 mg/m^2^ for paclitaxel (range: 44.4–200 mg/m^2^). During the 203 total GP cycles, the percentages of the planned day 8 and 15 treatment actually given were 70.4% for gemcitabine (143/203) and 59.6% for paclitaxel (121/203), repectively. Myelosuppression was the reason for most of the omitted treatments.

### 2.2. Treatment Efficacy

Table 2 shows the response rates to GP. The ORR was 35.8% (CR: 7.5%; PR: 28.3%), and the disease control rate was 69.8%. In terms of 1st-line therapy, the ORR was not significantly different in the two groups (MVAC: 57.8%; CG: 50.0%; *p* = 0.68). The ORR to GP was significantly higher in patients who responded to MVAC than in those who responded to GC (53.9% [14/26] vs. 0% [0/4], *p* = 0.045). The ORR to GP was nonsignificantly different in patients who did not respond to their 1st-line regimens (MVAC 21.1% [4/19] vs. GC 25.0% [1/4], *p* = 0.86).

The Kaplan–Meier analysis showed that, for 2nd-line chemotherapy, the median PFS was 7.3 months (range: 1.2–103.8 months) and the median OS was 12.3 months (range: 2.7–103.8 months), with the cumulative PFS and OS rates being 22.6% and 50.9% at 1 year, and 7.6% and 15.1% at 2 years (Figure 1A,B). A multivariate analysis adjusted for the effects of clinicopathologic factors showed that PS2 (hazard ratio [HR]: 6.73; 95% confidence interval [CI]: 2.64 to 17.23; *p* < 0.001), 1st-line non-response (HR: 1.92; 95% CI: 1.02 to 3.63; *p* = 0.044), and 2nd-line non-response (HR: 2.85; 95% CI: 1.37 to 5.94; *p* = 0.005) were independent risk factors for worse PFS; and PS2 (HR: 2.84; 95% CI: 1.19 to 6.81; *p* = 0.019) and 1st-line non-response (HR: 2.55; 95% CI: 1.12 to 5.77; *p* = 0.025) were independent risk factors for worse OS (Table 3 and Table 4). For 1st-line chemotherapy, the median PFS was 11.5 months (range: 1.2–125.3 months) and the median OS was 25.3 months (range: 6.0–125.3 months). The mean time to the first progression from the introduction of the 1st -line chemotherapy was not significantly different between the two regimens (GC: 16.0 months, MVAC: 14.0 months, *p* = 0.64). The mean OS from the introduction of the 1st-line chemotherapy was not significantly different between UTUC and BCa (UTUC: 25.2 months, BCa: 34.0 months, *p* = 0.16).

### 2.3. Adverse Events

Table 5 lists the hematologic and non-hematologic toxicities observed. Common hematologic AEs were neutropenia and anemia (both 88.7%). Grade 3 neutropenia occurred in 28 patients (52.9%), and grade 4 in 8 patients (15.1%). Febrile neutropenia was observed in 5 patients with grade 3 neutropenia (9.4%) and was cured with antibiotics. Four patients with grade 3 or greater anemia (7.5%) received a blood transfusion and experienced no severe secondary symptoms such as cardiovascular events. Peripheral sensory neuropathy and alopecia were frequently observed non-hematologic toxicities, but none were classified as greater than grade 3. No treatment-related deaths occurred in this study.

### 2.4. Predictive Role of PGK1

The expression of PGK1 (negative, low, and high) in representative immunostained tissues is shown in Figure 2A–C. Of the 28 patients eligible for the subgroup analysis of the predictive role of PGK1 in treatment response to CC, 5 (17.9%) had received GC and 23 (82.1%) had received MVAC as 1st-line treatment, and 16 (30.2%) had BCa specimens and 12 (22.6%) had UTUC specimens. As shown in Table 6, PGK1N was classified as LE in 12 (42.9%) and as HE in 16 (57.1%), and the non-response rate (stable + progressive disease) to 1st-line chemotherapy was significantly higher in patients with HE than in those with LE (HE: 69.7% [11/16]; LE: 16.7% [2/12]; *p* = 0.006). Based on the H-score, the receiver operating characteristic AUC for PGK1N in 1st-line chemotherapy was 0.78 (95% CI: 0.60 to 0.96). Using a cut-off of 35, PGK1N had a sensitivity of 84.6% and a specificity of 66.7% for predicting resistance to 1st-line chemotherapy. Non-response rates to GP were not significantly different in patients with HE and with LE (69.7% [11/16] and 58.3% [7/12], respectively). The expression of PGK1C was not related to treatment outcomes after both platinum-based chemotherapies and after GP.

## 3. Discussion

The present study, which includes a relatively large cohort compared with other studies of 2nd-line GP for aUC, updates our previous reports about this doublet [10,11,12,13]. In this cohort, GP was associated with an ORR of 35.8% and a median OS of 12.3 months with a tolerable toxicity profile. Multivariate analysis showed that non-response to GP was an independent predictor of worse PFS. Notably, patients who responded to MVAC compared with those who responded to GC responded to GP at a significantly higher rate. Additionally, PGK1N showed promise as a predictor of resistance to 1st-line platinum chemotherapy, having an AUC of 0.78 with a sensitivity of 84.6% and a specificity of 66.7%. We believe that our report is the first to evaluate the predictive role of PGK1 in treatment for UC. Furthermore, that predictive role was evident in two analyzed sites: nucleus and cytoplasm.

Treatment outcomes in the present cohort accord with those from several previous studies of 2nd-line GP in aUC, with the ORR ranging from 29.4% to 60.0%, the median PFS ranging from 4.3 to 6.4 months, and the median OS ranging from 11.3 to 14.4 months [10,11,12,13]. Although GP regimens can vary in dose and schedule, the reported ORR and median OS ranges can reasonably be interpreted as favorable, considering that in the KEYNOTE-045 trial, pembrolizumab was associated with an ORR of 21.1% and a median OS of 10.3 months [5]. Pembrolizumab is currently recognized as the standard 2nd-line treatment for mUC refractory to platinum-based chemotherapy. However, this ICI poses risks of severe AEs, which contraindicate it for patients with comorbidities such as interstitial pneumonia and autoimmune disease [5,6]. GP might therefore be an alternative to an ICI for such patients. Additionally, a subgroup analysis of IMvigor210 demonstrated satisfactory efficacy for atezolizumab regardless of prior lines of therapies for aUC. By sparing ICIs for later use, 2nd-line GP might contribute to broadening the treatment options for aUC [18].

Although more than double the number of patients received GP in the present study than in our last study, the rate and severity of AEs were acceptable [9]. Grades 3/4 neutropenia, febrile neutropenia, and grades 3/4 thrombocytopenia (hematologic AEs) occurred, respectively, in 52.8%/15.1%, 9.4% and 17.0%/0% of patients in the present study and in 46%/21%, 17% and 25%/4.2% of patients in the last study. No non-hematologic AE was classified as more than grade 3 in either study, and AEs were, overall, manageable, with no therapy-related death. Moreover, the AE profile in a recent retrospective study, which used almost the same 2nd-line GP regimen as in the present study but on an every-4-weeks schedule, were also in line with ours [13]. Not only were AEs during 2nd-line GP in the present study tolerable, but GP was also observed to be independently prognostic for PFS. Given that PFS commonly contributes to a better performance status, GP might also have advantages for outpatients.

Chemoresistance is a key problem in cancer treatment, and evidence increasingly indicates that treatment outcomes are poor after gemcitabine reuse. In vitro studies revealed that clusterin is involved in acquired gemcitabine resistance in BCa cells and pancreatic cancer cells by increasing Akt phosphorylation and cell survival [19]. In fact, the latest clinical study of 2nd-line GP found that prior use of gemcitabine was an independent prognostic factor for worse OS in aUC [13]. Our study supports those negative impacts of prior gemcitabine use on treatment outcomes given that the response to GP was significantly greater in responders to MVAC than in responders to GC responders, with only one patient of those who received GC responding to GP. Considering all the evidence, we believe that, for the initial treatment of aUC, a platinum-based regimen without gemcitabine should be selected, with GP being introduced at some later point during the treatment course.

Although the treatment sequence after the failure of 1st- and 2nd-line chemotherapy for aUC has been a stubborn clinical issue, enfortumab vedotin, an antibody–drug conjugate, has been widely approved as 3rd-line treatment thanks to the EV301 trial published in 2021 [20]. Pembrolizumab followed by enfortumab vedotin has since been the dominant treatment sequence for mUC refractory to 1st-line treatment. Intriguingly, however, two recent retrospective studies showed favorable results after reuse of CC after progression of mUC on an ICI or enfortumab vedotin, reporting a median OS of 9 months after an ICI and about 12 months after enfortumab vedotin. The mechanism at work might be a synergistic effect of CC in the post-ICI or post–enfortumab vedotin immunologic context [21,22]. For example, anti–PD-1 antibodies increase the number CD4+ cells, which normalizes tumor blood flow, allowing more drug to reach the tumor tissues, and long-term use of enfortumab vedotin makes tumor tissue sensitive to CCs [23,24]. Given the current standard treatment sequence of ICI and then enfortumab for mUC, a CC might be a good choice for 4th-line treatment. A biomolecular approach is mandatory for understanding the optimal treatment sequence for aUC.

Combined with clinical information, a biomolecular approach will help to facilitate a precision medicine approach for aUC. Recently, metabolic reprogramming proteins, especially PGK1, have been highlighted as an emerging hallmark of cancer, and our study had two striking findings in the analysis of PGK1 in aUC [25,26]. First, we separately evaluated the predictive role of PGK1 in cytoplasm and nucleus. PGK1 is generally considered to be localized on the cell membrane and in the cytoplasm, reflecting its well-known role of glycolysis [26]. Notably, however, some studies showed a translocation of PGK1 from the cytoplasm to the nucleus in advanced malignancies [27,28]. One study found that, in a brain tumor, PGK1N promoted DNA replication in cooperation with Cdc7 [27]. This reported PGK1N translocation might explain the association we observed here between high expression levels of PGK1N and a poor response to platinum chemotherapy and also the promise that PKG1N holds for predicting resistance to 1st-line chemotherapy. The AUC of 0.78 in the present study was favorable in comparison with other biomarkers considered to be predictive of the response to platinum-based chemotherapy for ovarian cancer (8 genes with an AUC ranging from 0.61 to 0.77) [29]. The overexpression of PGK1 has reportedly been associated with cisplatin resistance in ovarian and bladder cancer cells, which might be because PGK1 can induce multi-drug resistance through an MDR-1 independent pathway. [15,25,26,30,31] In the clinical use, as PGK1N had a sensitivity of 84.6% with a relatively weak possibility of false negative, the low expression of PGK1N might be helpful in identifying patients who can benefit from 1st-line platinum chemotherapy. Further studies are needed to reveal the exact mechanism of PGK1 in the acquisition of chemotherapeutic resistance in UC.

Our study has some limitations. First, the study’s retrospective design and its lack of randomization might have introduced bias into the patient selection process. Second, aspects of the 1st-line platinum-based chemotherapy such as treatment intensity were decided by the doctor in charge in each case, and those differences might have influenced our results. Third, although the sample size in the study was small, it was comparable to those in previous studies, given that a patient’s symptom burden from disease progression after 1st-line treatment often contraindicates 2nd-line chemotherapy [10,11,12,13]. Fourth, we did not evaluate the genetic impact of UTUC and BCa on the clinical outcomes. However, OS from the introduction of the 1st-line chemotherapy was not significantly different between the two tumor locations. Fifth, we were not able to evaluate the expression of PGK1 in all 53 specimens. In some cases, the quality of preservation was poor or the specimen was small; in other cases, patients were diagnosed using urinary cytology and no pathology specimen was available. However, the predictive value of PGK1 for 1st-line treatment in aUC is noteworthy, given the scant data.

## 4. Materials and Methods

### 4.1. Patients

The study was approved by the Institutional Review Board of Kitasato University School of Medicine and Hospital and was conducted in accordance with the Declaration of Helsinki (B17-010, B19-192). Potential participants received information on the opportunity to opt out via our website and posters. We retrospectively reviewed clinical data for 69 patients who received GP as 2nd-line chemotherapy for aUC, including locally advanced (T2–4, N+) or metastatic (M1) BCa and UTUC at Kitasato University Hospital between 2005 and 2021. Patients with a history of peptide therapy (*n* = 5) and those who received only 1 cycle of GP (*n* = 11) were excluded from the study.

Patients were required to have an Eastern Cooperative Oncology Group performance status of 2 (PS2) or lower. In each patient, a histopathologic diagnosis was made by either surgical treatment, biopsy of the primary lesion, or urine cytology. All patients had previously received platinum-based chemotherapies of either combination methotrexate–vinblastine–doxorubicin–cisplatin (MVAC) or the gemcitabine–cisplatin doublet (GC). Patients were eligible if their disease had progressed to advanced or metastatic disease at any time after 1st-line therapy or within 12 months of neoadjuvant or adjuvant chemotherapy.

### 4.2. Chemotherapy Regimen

The GP regimen delivered paclitaxel 200 mg/m^2^ on day 1 and gemcitabine 1000 mg/m^2^ on days 1, 8 and 15. The treatment course was repeated every 3 weeks. On the first day of each course, full doses of both drugs were given if the patient’s white blood cell count was greater than 3000/mL and the platelet count was greater than 100,000/mL. If those counts were lower than those cut-offs, treatment was delayed for 1 week. On days 8 and 15 of each cycle, full-dose gemcitabine was given if the patient had a white blood cell count greater than 3000/mL and a platelet count greater than 75,000/mL. These treatment plans were consistent with those in our previous reports [10,11].

### 4.3. Treatment Evaluation

Tumors were assessed by computed tomography or magnetic resonance imaging every two cycles, and responses were determined at least 4 weeks after administration. Data on patient characteristics collected from medical charts included age at the introduction of GP, sex, performance status, primary organ, types of surgical management, types of 1st-line platinum-based chemotherapy, recurrence sites, types of AEs, response to 1st-line platinum-based chemotherapy and to GP, and time to recurrence and death from initiation of 1st-line platinum-based chemotherapy and of GP. Tumor response was defined according to the Response Evaluation Criteria in Solid Tumors, version 1.1. AEs were classified according to the Common Terminology Criteria for Adverse Events, version 4.0. OS and progression-free survival (PFS) were calculated from the first date of 1st-line platinum-based chemotherapy and of GP to the date of the last follow-up or death from any cause or from disease progression, respectively.

### 4.4. Immunohistochemistry and Scoring of PGK1

Formalin-fixed, paraffin-embedded tissue blocks representing the most invasive areas of each tumor were collected for further investigation. Sections of 3 µm were immunostained using the Bond-Max automated immunohistochemistry system and Bond Polymer Refine Detection kit (DC 9800: Leica Biosystems, Newcastle, UK). Sections were deparaffinized and pretreated with Bond Epitope Retrieval Solution 2 (Leica Biosystems, Newcastle, UK) at 100 °C for 20 min. After washing and peroxidase blocking for 10 min, specimens were re-washed, and immunohistochemistry for PGK1 was performed as follows. Sections were incubated first with rabbit polyclonal anti-PGK1 antibody (PAB2618: Abnova, Taipai, Taiwan; diluted 1:400) for 30 min and then with EnVision FLEX+ Rabbit Linker (Dako, Glostrup, Denmark) for 15 min. Finally, the sections were incubated with Bond Polymer (Leica Biosystems, Newcastle, UK) for 10 min, developed with 3,30-diaminobenzidine chromogen for 10 min, and counterstained with hematoxylin for 5 min. Sections treated with Bond Primary Antibody Diluent (Leica Biosystems, Newcastle, UK) replacing the primary antibody were used as negative controls.

PGK1 was located in the cytoplasm and/or plasma and nuclear membrane of tumor cells. Expression levels of PGK1 were assessed with respect to PGK1 in the nucleus (PGK1N) and PGK1 in the cytoplasm (PGK1C). Specimen used in the analysis had been obtained by curative-intent surgery, including transurethral resection of bladder cancer, partial cystectomy, radical cystectomy or radical nephroureterectomy, and by transurethral resection of bladder cancer for diagnostic purposes. PGK1 expression levels were scored using the following scale: 0, no staining; 1, weak staining; 2, moderate staining; and 3, strong staining. The percentage of tumor cells expressing PGK1 was calculated and multiplied by the staining scores over an average of three areas to obtain a semiquantitative H-score (maximum value: 300). All immunostained sections were reviewed by 2 investigators (D.K., Y.S.) who had no knowledge of the clinical data. Discordant cases were reviewed and discussed until consensus was reached. A PGK1 score of <35 was categorized as low expression (LE) and that of ≥35 as high expression (HE). Finally, associations of treatment outcomes after 1st- and 2nd-line chemotherapy with the PGK1N and PGK1C expression levels were examined.

### 4.5. Statistical Analysis

Comparisons of response rates to 1st- and 2nd-line chemotherapy were evaluated using the chi-squared test (or the Fisher exact test, if appropriate). OS and PFS after the first date of GP were estimated by the Kaplan–Meier method with log-rank test. A multivariate analysis for OS and PFS was performed using a Cox proportional hazards regression model, controlling for the effects of patient characteristics. The area under the curve (AUC) and best cut-off point were calculated using a receiver operating characteristic analysis. All statistical analyses were performed in the Stata software application (version 13 for Windows: StataCorp LP, College Station, TX, USA). All *p* values are 2-sided, and *p* < 0.05 was considered statistically significant.

## 5. Conclusions

To summarize, the present study demonstrates the efficacy and tolerability of 2nd-line GP for patients with aUC refractory to platinum-based chemotherapy. Moreover, we are the first to report PGK1N as a valuable predictor of resistance to platinum-based chemotherapy in aUC. Hence, 2nd-line GP is a promising option even in the era of ICIs, and PGK1N might be of great help in facilitating a precision medicine approach to aUC.

## Figures and Tables

**Figure 1 ijms-23-12119-f001:**
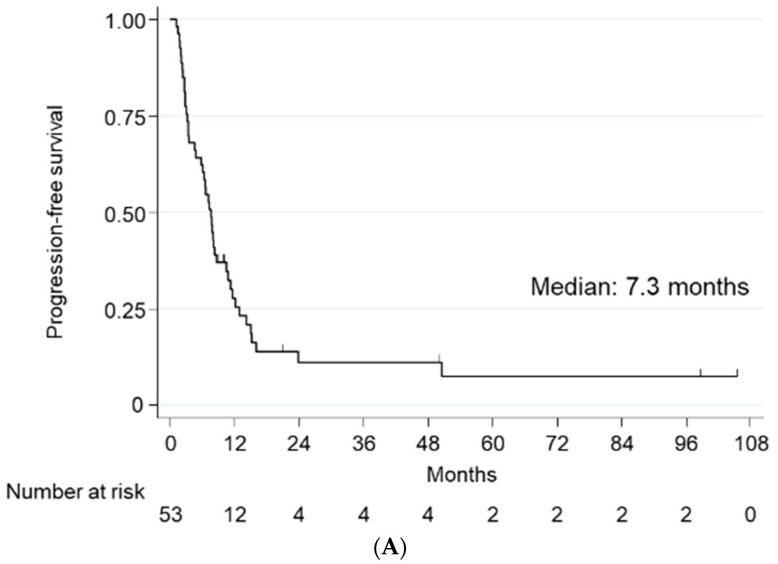
Kaplan–Meier estimates of (**A**) progression-free survival and (**B**) overall survival.

**Figure 2 ijms-23-12119-f002:**
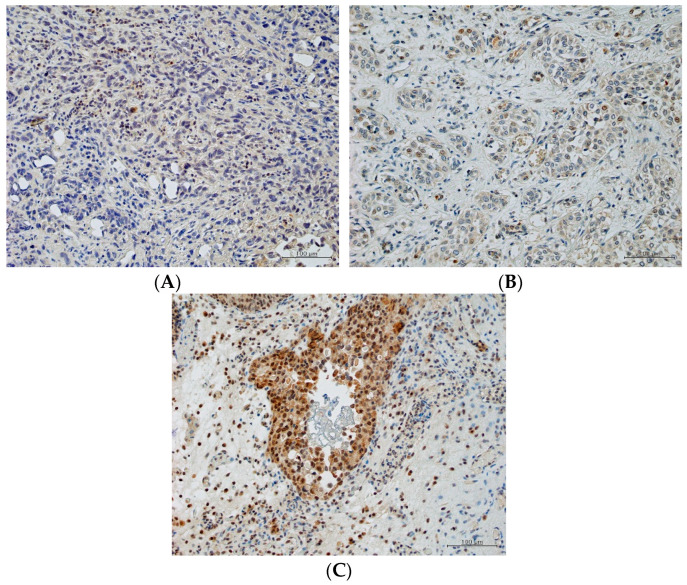
Examples of (**A**) negative, (**B**) low, and (**C**) high PGK1 expression by immunostaining of specimens obtained by curative-intent surgery or transurethral resection of bladder cancer for diagnostic purposes. All 200× original magnification.

**Table 1 ijms-23-12119-t001:** Patient characteristics.

Characteristic	Value
Age (years)	
Median	68
IQR	64–74
Sex (N [%])	
Male	42 (79.1)
Female	11 (20.8)
Performance status (N [%])	
0	20 (37.3)
1	26 (49.1)
2	27 (13.2)
Primary organ (N [%])	
Bladder	32 (60.4)
Upper urinary tract	21 (39.6)
Surgical management (N [%])	
Radical cystectomy	7 (13.2)
Partial cystectomy	2 (3.8)
Radical nephroureterectomy	14 (26.4)
Transurethral resection of bladder cancer	18 (33.9)
Lymph node dissection	2 (3.8)
Inoperable	10 (18.9)
1st-line chemotherapy regimen (N [%])	
MVAC	45 (84.9)
GC	8 (15.1)
Metastatic sites (N [%])	
Lymph node	24 (42.1)
Lung	20 (35.1)
Bone	6 (10.5)
Liver	4 (7.0)
Peritoneum	2 (3.5)
Adrenal	1 (1.8)

MVAC, methotrexate–vinblastine–doxorubicin–cisplatin; GC, gemcitabine-cisplatin.

**Table 2 ijms-23-12119-t002:** Tumor response to gemcitabine and paclitaxel therapy by chemotherapy regimen.

Response Type	Patients (N [%])	Responded to Chemotherapy Used in 1st Line (N [%])					
		Yes			No		
		MVAC	GC	*p* Value	MVAC	GC	*p* Value
Complete response (CR)	4 (7.5)	14 (53.9)	0 (0)	0.045	4 (21.1)	1 (25.0)	0.86
Partial response (PR)	15 (28.3)						
Stable disease (SD)	18 (34.0)	12 (46.1)	4 (100)		15 (78.9)	3 (75.0)	
Progressive disease (PD)	16 (30.2)						
Overall response rate (CR+PR)	19 (35.8)						
Disease control (CR+PR+SD)	37 (69.8)						

MVAC, methotrexate–vinblastine–doxorubicin–cisplatin; GC, gemcitabine-cisplatin.

**Table 3 ijms-23-12119-t003:** Univariate and multivariate analyses for progression-free survival.

Variable	Comparator	Univariate Analysis			Multivariate Analysis		
		HR	95% CI	*p* Value	HR	95% CI	*p* Value
Age	≥70	0.99	0.56 to 1.75	0.43	0.92	0.50 to 1.69	0.78
	≤69	1.0			1.0		
Performance status	2	6.16	2.55 to 12.83	<0.001	6.73	2.64 to 17.2	<0.001
	≤1	1.0			1.0		
Visceral metastasis	Present	0.92	0.53 to 1.58	0.88	1.01	0.54 to 1.86	0.99
	Absent	1.0			1.0		
Responded to 1st-line chemotherapy	No	2.07	1.17 to 3.68	0.013	1.92	1.02 to 3.63	0.044
	Yes	1.0			1.0		
Responded to GP	No	3.10	1.61 to 5.97	0.006	2.85	1.37 to 5.94	0.005
	Yes	1.0			1.0		

HR, hazard ratio; CI, confidence interval; GP, gemcitabine–paclitaxel.

**Table 4 ijms-23-12119-t004:** Univariate and multivariate analyses for overall survival.

Variable	Comparator	Univariate Analysis			Multivariate Analysis		
		HR	95% CI	*p* Value	HR	95% CI	*p* Value
Age	≥70	0.94	0.49 to 1.82	0.86	0.51	0.25 to 1.04	0.07
	≤69	1.0			1.0		
Performance status	2	4.08	1.82 to 9.16	0.004	2.84	1.19 to 6.81	0.019
	≤1	1.0			1.0		
Visceral metastasis	Present	0.91	0.48 to 1.7	0.92	0.80	0.39 to 1.64	0.54
	Absent	1.0			1.0		
Response to 1st-line chemotherapy	No	1.40	0.71 to 2.81	0.33	2.55	1.12 to 5.77	0.025
	Yes	1.0			1.0		
Response to GP	No	2.59	1.19 to 5.65	0.017	1.57	0.70 to 3.56	0.28
	Yes	1.0			1.0		

**Table 5 ijms-23-12119-t005:** Toxicities with gemcitabine–paclitaxel.

Adverse Event	Grade (N [%])				
	1	2	3	4	All
Neutropenia	4	7	28	8	47 (88.7)
Anemia	17	16	11	3	47 (88.7)
Thrombocytopenia	14	7	9	—	30 (56.6)
Febrile neutropenia	—	—	5	—	5 (9.4)
Neuropathy	11	10	—	—	21 (39.6)
Skin rash	1	3	—	—	4 (7.5)
Nausea/vomiting	7	1	—	—	8 (15.1)
Alopecia	21	1	—	—	22 (41.5)
Liver dysfunction	9	1	—	—	10 (18.9)

**Table 6 ijms-23-12119-t006:** Characteristics of the patients eligible for the subgroup analysis of the predictive role of PGK1.

Characteristics (N [%])	LE (*n* = 12)	HE (*n* = 16)	*p* Value
Primary organ			
Bladder	7 (58.3)	9 (56.3)	0.91
Ureter	5 (41.7)	7 (43.7)	
1st-line chemotherapy			
MVAC	10 (83.3)	13 (76.9)	0.89
GC	2 (16.7)	3 (23.1)	
Non response			
1st-line chemotherapy	2 (16.7)	11 (69.7)	0.006
2nd-line chemotherapy	7 (58.3)	11 (69.7)	0.57

LE, low expression; HE, high expression; MVAC, methotrexate–vinblastine–doxorubicin–cisplatin; GC, gemcitabine-cisplatin.

## Data Availability

The datasets used and/or analyzed during the present study are available from the corresponding author on reasonable request.

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
