# Peer review of "Gemcitabine–Paclitaxel Chemotherapy for Patients with Advanced Urothelial Cancer Refractory to Cisplatin-Based Chemotherapy: Predictive Role of PGK1 for Treatment Response to Cytotoxic Chemotherapy"

_ijms, 2022, doi:10.3390/ijms232012119_

Round 1
Reviewer 1 Report
1. Different 1st line treatments may provoke the different responses of cancer cells. How can the authors avoid this discrepancy? Also, UTUCand UBUC might have different gene signatures. It is hard to conclude GP treatment in this study.
2. Figure 2 is missing.
3. Make a table for section 2.4.
4. The survival median for PFS and OS is 7.3 months and 12.3 months, respectively. The efficacy is not good.
5. The patient number is too low to conclude GP efficacy.
6. At a cut-off of 35, 66.7% specificity is unsuitable for prognosis.
7. The authors may use various UC cell lines to validate the PGK1N expression in UC cell lines and study the PGK1N-involved critical mechanism.
Author Response
September 26th, 2022
Professor Maurizio Battino,
Editor-in-chief, International Journal of Molecular Sciences
Dear Professor Maurizio Battino,
RE: ID ijms-1906448Title: " Gemcitabine–paclitaxel chemotherapy for patients with advanced
urothelial cancer refractory to cisplatin-based chemotherapy: predictive role
of PGK1 for treatment response to cytotoxic chemotherapy"
Thank you for the critical review of our manuscript and the opportunity to submit revisions. We have reviewed the comments from the two reviewers and made revisions where we feel appropriate. A list of the response to each of the comments were written in the attached word file.

Reviewer 2 Report
The manuscript is well written, and the research well-conducted.
The research problem is relevant and insightful, the rational for studying the problem was well stated.
There is an adequate introductory or background, there is recognition of gaps in knowledge and in the literature, and the relevance of the significance of other current research on the subject was highlighted. The authors adequately summarized the relevant literature and the references were relevant to the paper.
The research questions were well-stated, the interpretation of study findings was appropriate, the discussion well-conducted.
there was an appropriate use and interpretation of statistics and the statistical and qualitative data presentation was clear enough.
The results could be potentially reproducible
However, there are a lot of limitations: retrospective design, lack of randomization, the 1st-line platinum-based chemotherapy decided by the doctor in charge in each case, the small sample size, the inability to evaluate the expression of PGK1 in all 53 specimens, a control group is lacking; moreover, the patients were consecutive?
In my opinion the authors might improve all these aspects, by tailoring the study design.
Author Response

(The authors gave the same response as above.)

Round 2
Reviewer 1 Report
1. The authors do not explain how their studies can circumvent the different gene signatures between UTUC and UBUC.
2. The responses to questions 1, 4, 5, and 6 may be inserted into the text.
3. At least the authors may use western immunoblotting to measure the translocation of PGK1 from the cytoplasm to the nucleus in UBUC cell lines with different grades.
Author Response
October 4th, 2022
Professor Maurizio Battino,
Editor-in-chief, International Journal of Molecular Sciences
Dear Professor Maurizio Battino,
RE: Manuscript ID: ijms-1906448Title: " Gemcitabine–paclitaxel chemotherapy for patients with advanced
urothelial cancer refractory to cisplatin-based chemotherapy: predictive role
of PGK1 for treatment response to cytotoxic chemotherapy"
Thank you for the critical review of our manuscript and the opportunity to submit the second revision. We have reviewed the comments from the one reviewer and made revisions where we feel appropriate. A list of the response to each of the comments as follows.
Reviewer: 1
#1. 1. The authors do not explain how their studies can circumvent the different gene signatures between UTUC and UBUC.
Answer: Thank you very much for sharp opinion. As I have answered in the last revision (1st round), we tried to make relatively homogenous cohort based on the clinical background. Furthermore, the clinical outcome in the present study between UTUC and UBUC was not significantly different (such as the mean overall survival from the introduction of the 1st line chemotherapy). However, I totally agree with your comment that we did not directly evaluate genetic background of UTUC and UBUC, and in fact, some literatures recently showed the difference in chemosensitivity between UTUC and UBUC, potentially due to the molecular divergence among the two types of cancers [1,2]. We therefore include this point in the limitation section and will analyze such genetic impact on the clinical outcomes in the next issue.
Page4 “Results”, line107-108.
The mean OS from the introduction of the 1st-line chemotherapy was not significantly different between UTUC and BCa (UTUC: 25.2 months, BCa: 34.0 months, P = 0.16).
Page9 “Discussion”, line249-252.
Fourth, we did not evaluate the genetic impact of UTUC and UBUC on the clinical outcomes. However, OS from the introduction of the 1st-line chemotherapy was not significantly different between the two tumor locations.
#2. The responses to questions 1, 4, 5, and 6 may be inserted into the text.
Answer: Thank you very much for your kind suggestion.
â‘ We put following sentences in the main manuscript accordingly.
Page4 “Results”, line105-107.
The mean time to the first progression from the introduction of the 1st-line chemotherapy was not significantly different between the two regimens (GC: 16.0 months, MVAC: 14.0 months, P = 0.64, not shown in the main manuscript).
Page4 “Results”, line107-108.
The mean OS from the introduction of the 1st-line chemotherapy was not significantly different between UTUC and BCa (UTUC: 25.2 months, BCa: 34.0 months, P = 0.16).
Page8 “Discussion”, line238-239.
As PGK1N had a sensitivity of 84.6% with a relatively weak possibility of false negative,
â‘¡We have already answered questions from 4 to 6 in the main manuscript as below:
Question 4→Page7 “Discussion”, line168-193 (paragraph 2 and 3).
Question 5→Page8 “Discussion”, line247-249.
Question 6→Page8 “Discussion”, line232-235.
#3. At least the authors may use western immunoblotting to measure the translocation of PGK1 from the cytoplasm to the nucleus in UBUC cell lines with different grades.
Answer: We appreciate for your suggestion. Western immunoblotting in the cell lines with different grades will be fascinating because it will be of great help in exploring the mechanism of PGK1 and verifying such translocation of PGK1. In our previous study, western immunoblotting was performed using different grades of bladder cancer cell lines including T24, T24CDDPR (cisplatin resisted cell), RT4, RT4ρ0, 5637, EJ, and TCCSUP (3, current manuscript reference number 15). There were slight or none expressions of PGK1 in western immunoblotting, except for T24 and T24CDDPR. In addition, T24CDDPR showed high expression of PGK1 compared to T24 in the same experiment. According to these results, we conducted the present study. On the other hand, we think western immunoblotting itself is relatively difficult to measure the translocation of PGK1 from the cytoplasm to the nucleus in UBUC cell lines, because cell lines were homogenized and extracted proteins before the experiments. Currently, in order to overcome these limitations, we have started a new project, in which the predictive value of PGK1 in patients with neoadjuvant cisplatin-based chemotherapy followed by radical cystectomy are evaluated, especially for the further understanding of the association between chemosensitivity and PGK1 in UBUC. Thank you again for your witty advice.
Reference
- Robertson, G.; Kim, J.; Ahmadie, H.A.; Bellmunt, J.; Guo, G.; Cherniack, A.D.; et al. Comprehensive molecular characterization of muscle-invasive bladder cancer. Cell. 2018, 174, 1033.
- Li, X.; Li, S.; Chi, Z.; Cui, C.; Si, L.; Yan, X.; et al. Clinicopathological characteristics, prognosis, and chemosensitivity in patients with metastatic upper tract urothelial carcinoma. Urol Oncol. 2021, 39, e1-75.
- Taoka, Y.; Matsumoto, K.; Ohashi, K.; Minamida, S.; Hagiwara, M.; Nagi, S.; et al. Protein expression profile related to cisplatin resistance in bladder cancer cell lines detected by two-dimensional gel electrophoresis. Biomed Res. 2015, 36, 253-261.
Dai Koguchi, MD, PhD.
Kazumasa Matsumoto, MD, PhD.
Department of Urology Kitasato University School of Medicine 1-15-1
Kitasato Minami-ku, Sagamihara, Kanagawa 252-0374, Japan
TEL: +81-42-778-9091 FAX: +81-42-778-9374 E-mail: kazumasa@cd5.so-net.ne.jp

Round 3
Reviewer 1 Report
No comments